# Risk Factors for Same Pathogen Sepsis Readmission Following Hospitalization for Septic Shock

**DOI:** 10.3390/jcm8020181

**Published:** 2019-02-03

**Authors:** June-sung Kim, Youn-Jung Kim, Seung Mok Ryoo, Chang Hwan Sohn, Shin Ahn, Dong Woo Seo, Kyoung Soo Lim, Won Young Kim

**Affiliations:** Department of Emergency Medicine, University of Ulsan College of Medicine, Asan Medical Center, 88, Olympic-ro 43-gil, Songpa-gu, Seoul 05505, Korea; jsmeet09@gmail.com (J.-s.K.); yjkim.em@gmail.com (Y.-J.K.); chrisryoo@naver.com (S.M.R.); schwan97@gmail.com (C.H.S.); ans1023@gmail.com (S.A.); leiseo@gmail.com (D.W.S.); kslim@amc.seoul.kr (K.S.L.)

**Keywords:** septic shock, hospital readmission, outcomes, risk factors, survivor

## Abstract

(1) Background: Septic shock survivors frequently readmit because of subsequent infection. This study aimed to determine the rate and risk factors for same pathogen sepsis readmissions following hospitalization for septic shock. (2) Methods: We performed this retrospective study using data from a prospective septic shock registry at a single urban tertiary center. All the patients were treated with a protocol-driven resuscitation bundle therapy between 2011 and 2016. We collected data from adult (older than 18 years) patients readmitted with sepsis within 90 days of discharge following hospitalization for septic shock. (3) Results: Among 2062 septic shock patients, 690 were readmitted within 90 days of discharge. After excluding scheduled and non-sepsis admissions, we analyzed the data from 274 (13.3%) patients readmitted for sepsis. Most of the readmissions following septic shock were new infections rather than relapses of the initial infection. The culture-negative rate was 51.4% (141/274), while the same pathogen was isolated in 25% of cases (69/274). Multivariate analysis revealed that previous gram-negative bacteremia (OR, 9.902; 95% CI, 2.843–34.489), urinary tract infection (OR, 4.331; 95% CI, 1.723–10.882) and same site infection (OR, 6.894; 95% CI, 2.390–19.886) were significantly associated with readmission for sepsis caused by the same pathogen. (4) Conclusions: The sepsis readmission rate following the previous hospitalization for septic shock was 13.3% and one-quarter of those patients had the same pathogen isolated. Previous gram-negative bacteremia, and/or same site infection are predisposing factors for recurrent same-pathogen sepsis.

## 1. Introduction

The incidence of sepsis has been increasing at the national level and the condition remains a considerable economic burden [1,2,3]. Since the establishment of the management guidelines and quality-improvement program for sepsis by the Surviving Sepsis Campaign, a global initiative aimed at improving survival in patients with sepsis and septic shock, the sepsis-related in-hospital mortality appears to be decreasing [4]. However, patients with sepsis who discharged from hospitalization continue to face an increased risk of readmission and the costs of hospital readmissions after sepsis have been estimated at 3 billion dollars [5]. Moreover, the cumulative mortality after severe sepsis admission increases continuously from less than 30% at one month, to approximately 40% at 90 days and to more than 60% after 5 years [6].

Although sepsis survivors tend to suffer from subsequent infections and readmission within the following year (particularly during the first 90 days after discharge), it is unclear whether the repetitive infections are from relapsed/persistent infections or from new ones [7,8]. Moreover, risk factors for same-pathogen sepsis readmissions following hospitalization for sepsis have not been reported. This information would be useful to help physicians choose an empirical antibiotic (to decide whether to use the same antibiotics based on previous culture results or different broad-spectrum antibiotics) [6,9].

Our main purpose with this study was to determine the extent to which 90-day readmissions for recurrent sepsis were due to the same infections and to identify the predictors for infections caused by the same organisms.

## 2. Materials and Methods

### 2.1. Setting and Study Population

We conducted this retrospective cohort analysis of data from a prospective data registry at an urban academic adult emergency department, within a tertiary referral center in the Republic of Korea, with annual patient admissions of more than 110,000 patients. The Institutional Review Board of our hospital approved the study (Study No. 2016-0548) and waived the requirement for informed consents because the study involved the analysis of a case registry records. We included data from all adult patients (≥18 years) with septic shock diagnosed in the emergency department and treated with protocol-driven resuscitation bundle therapy and whose data had been prospectively collected in our institution’s Septic Shock Registry. We used a definition of septic shock involving refractory hypotension (mean arterial pressure ≤65 mm Hg) requiring vasopressors despite adequate fluid therapy or a blood lactate concentration of at least 4 mmol/L, according to previous definitions [10]. We excluded data from patients with a “do not attempt resuscitation” status and from those who were transferred to another hospital during the initial resuscitation. The registry contains demographic, clinical and laboratory data, as well as clinical outcomes. In the end, we analyzed data from the records on septic shock patients readmitted within 90 days of discharge between January 2011 and December 2016. We excluded patients who revisited because of scheduled chemotherapy, operations, interventions and routine work-ups and we categorized 90-day readmissions as being either due to sepsis or to another reason. To minimize missing sepsis cases, we checked both admission and discharge notes, including the final diagnoses.

### 2.2. Data collection and Definition

According to the surviving sepsis campaign protocol, microbial studies were conducted within 3 h of septic shock recognition [11]. Blood cultures were obtained from samples at two or more different anatomical sites according to standard practice [12]. When an indwelling catheter was present, one blood sample was obtained through the catheter and the rest were taken from different peripheral venous sites. Site specific cultures such as sputum, urine, pus and stool were obtained according to clinician judgement and local practice. We categorized microorganisms, antigens and toxins according to culture results as follows: bacteria (gram-positive, gram-negative, polymicrobial), *Clostridium difficile*, viral, fungal or culture negative in blood, urine, sputum and pus; *Legionella pneumophila* and *Streptococcus pneumoniae* urine antigen; and *Clostridium difficile* toxin assay of stool. We grouped infection sites as urinary, gastrointestinal, hepatobiliary, pulmonary, skin and soft tissue, blood stream and central nervous system. We considered fevers without a definite focus as unknown sites.

When the “same organism” was isolated upon readmission that had been initially isolated, we assigned the cases to the “same organism” group and when it was not, we assigned them to the “different organism” group. When all the cultures were negative (including those in the first and second admissions) reflecting no definite growth of any specimen, we included these cases into the “unknown organism” group. In addition, we included primary bacteremia or febrile neutropenia without definitive foci into the “other site” group.

### 2.3. Statistical Analyses

The SPSS Statistics for Windows software version 23 (SPSS, Chicago, IL, United States) was used to perform all the statistical analyses. We expressed all continuous variables as medians and interquartile ranges. The Kolmogorov-Smirnov test was used to examine the normality of distribution. The Kruskal-Wallis test was used for comparisons between same, different and unknown sepsis organism groups. We analyzed categorical variables with 2-tailed chi-squared or with Fisher’s exact tests. To identify risk factors for the identical pathogen, we included variables with an entry-level significance of *p* < 0.05 in the univariate analysis in a stepwise multivariate analysis and after using the process of forward-backward stepwise regression, we reported the results as odds ratios (ORs) and 95% confidence intervals (CIs). In patients with multiple admission-required infections, we used only the first infection. In addition, standard statistical methods were used to calculate sensitivity, specificity, positive predictive value (PPV), negative predictive value (NPV) and positive and negative likelihood ratio (PLR, NLR). We considered *p*-values lower than 0.05 as statistically significant.

The main clinical outcome was finding the rate and risk factors of readmission due to sepsis caused by the “same organism” within 90 days of discharge following treatment for septic shock.

## 3. Results

### 3.1. Baseline Characteristics

From the registry, we analyzed the data for 2062 septic shock patients, of whom, 690 were readmitted within 90 days of discharge. We excluded 297 patients with non-sepsis related readmissions and 119 patients with scheduled readmissions (44 chemotherapy, 32 operation, 27 routine stents changed/removed and 15 routine work ups); and, in the end, used the data from 274 (13.3%) patients with sepsis readmission. Of the 274 cases, 69 (25%) were in the “same organism” group (Figure 1).

Table 1 presents the baseline characteristics of patients who were readmitted due to sepsis. We found no age-related differences between the three groups. Male was more common in unknown group. Malignancy was the most common underlying disorder followed by hypertension. We did not observe significant differences in previous medical conditions such as hypertension, stroke, diabetes, coronary artery disease, chronic pulmonary disease, liver cirrhosis, chronic renal failure or malignancy between the groups. Over the entire study period, 28- and 90-day mortalities, mechanical ventilation requirements and length of intensive care unit stays were statistically similar in three groups. However, we found the same site of reinfection to be significantly more frequent in the “same organism” group than in the “different organism” and “unknown organism” group (89.9% vs. 57.8% vs. 69.5%; *p* < 0.001).

### 3.2. Etiology of Recurrent Sepsis

The most common category of organism causing repetitive sepsis was the gram-negatives (*n* = 180; 65.7%), meanwhile, gram-positive infections were not as common (*n* = 34; 12.4%) (Table 2). Identical gram-negative species caused recurrent sepsis were predominant in same group (95.7% vs. 71.8% vs. 48.9%, respectively). The infection sites in cases of sepsis readmission included hepatobiliary (37.6), gastrointestinal (15.7%), respiratory (26.3%) and urinary tract (17.1%). Culture-negative samples comprised 141 (51.4%) of the total and the respiratory tract (*n* = 44; 31.2%) and intraabdominal system (*n* = 146; 53.3%) were common. In urinary system, same pathogen was isolated more often than others (34.8% vs. 20.3% vs. 7.1%, respectively). Table 3 presents the frequencies of microbes causing recurring sepsis. *Escherichia coli* was the most frequently grown gram-negative species (*n* = 31; 72%) followed by *Klebsiella spp.* (*n* = 7; 16%).

### 3.3. Risk Factors Independently Associated with Sepsis due to the Same Organism

The candidate risk factors associated with recurrent sepsis due to the “same organism” included the following: gender, same site, gram-positive, gram-negative, urinary, gastrointestinal, hepatobiliary and respiratory tract infections (Table 4). In the multivariable model, three clinical factors were independent risk factors for infections by the same organism: same site (OR, 6.894; 95% CI, 2.390–19.886; *p* < 0.001), gram-negative (OR, 9.902; 95% CI, 2.843–34.489; *p* < 0.001) and UTI (OR, 4.331; 95% CI, 1.723–10.882; *p* = 0.002).

To determine the predictive characteristics of these three parameters, we calculated sensitivity, specificity, PPV, NPV, PLR and NLR (Table 5). Among these, the gram-negative variable had the highest sensitivity (95.65%) and NPV (96.81%). We calculated additional variables including combinations of two factors. When all two of the parameters were included, the specificity increased to 58.4%.

## 4. Discussion

Our results revealed that (1) readmissions due to sepsis following hospitalization for septic shock were common; (2) one in four readmissions were due to recurrent sepsis by the same organism; (3) the most common same pathogens were gram-negative, particularly *E. coli*, with the predominant site of infection being the intraabdominal system; and, (4) previous gram-negative bacteremia and same site infection were independent predictors for same pathogen sepsis readmission following hospitalization for septic shock.

Our results confirm those of earlier studies which reported that sepsis patients get readmitted because of infection: Prescott et al. found that around 40% of severe sepsis survivors were re-hospitalized within 90 days with rates of readmission for sepsis being higher than those of other acute medical conditions such as congestive heart, acute renal and respiratory failures [13]. Ortego et al. also reported the most common cause of rehospitalization as infection including cellulitis, pneumonia and urinary tract infections [14]. Malignancy, length of stay, initial serum lactate levels and high APACHE II scores were risk factors associated with 30-day readmission [14]. Both of the studies focused on readmission itself; however, the authors did not characterize the specific causative organisms.

Repeat infections are due to new organisms or same persistent or relapsing organisms. We hypothesize that differentiating same or new infections is important because the initial treatment strategy for eliminating the causative organism such as duration and type of antibiotics usage depends on this knowledge. Among 69 same organism patients, 9 pathogens (13%; *E. coli* = 5; *Klebsiella species* = 2; *Enterobacter* = 1; *Pseudomonas* = 1) became resistant and rest pathogens had same susceptibilities of previous antibiotics. However, it cannot be conclusive for providing guideline of empirical antibiotics because of relatively small sample size in our study. Sun et al. reported that half the cases were categorized as recurrent/unresolved infections whereas half were due to the same pathogen [15]. In a recent study, DeMerle et al. reported that only one fifth of infections were confirmed to be caused by persistent “same organisms” and that the majority were caused by new infections [16]. This large difference gap between the two studies may be due to differences in dealing with culture negative cases. Because of the uncertainty, we considered the culture negative as “different organism” group for the sensitive analysis and the trend of the results were similar (Appendix A) and with this approach we observed 75% new, different infections and 25% same persistent or relapsing pathogens. With the increasing use of culture-independent methods, it may be possible to clarify the etiology of culture negative infections [9].

Studies have demonstrated that sepsis induces an acquired immunosuppression by activating complex proinflammatory and anti-inflammatory mechanisms [17]. Such immune deficiency makes it difficult to eradicate the primary infection or predisposes patients to exposure by new infections [18]. We postulate that different types of organism have distinct patterns of subsequent infection. Different, new gram-positive and viral species invade the host, while new gram-negative bacteria tend to induce subsequent sepsis. A study reported that *E. coli* can persist in the fecal flora after elimination from the urinary tract and may, subsequently, recolonize the urethra and bladder and cause recurrent urinary tract infections [19]. Reports have shown that repeat gram-negative bacterial infections are common largely due to reduced antibiotic susceptibility patterns as compared to prior infections [20]. Future work should identify risk factors for resistance-related pathogens and antibiotic types.

Moreover, our results suggest that a high proportion of sepsis readmissions are due to infections at the same site. This may be caused by the regional effect of sepsis, including both microbiome and anatomic disruptions, allowing easy re-entry of infectious organisms [21,22]. These results are in close agreement with those of the research by DeMerle et al. [16]. They postulated that hypoperfusion and subsequent reperfusion of the intestine cause bacterial translocation and indwelling catheters and invasive procedures predispose patients to recurrent sepsis [16,23]. In our study, hepatobiliary site infections, mostly cholangitis, were predominant and recurrent infections may have been caused by recurrent invasive procedures for drainage.

We are aware of the limitations in our study. First, this study is from a single institution, which limits the generalization of our findings to other populations because of the local sepsis patterns. Second, because of the study period, we relied on the explicit Sepsis-2 definition to identify and enroll sepsis and septic shock cases to the registry. Thus, it is possible that we may have missed some cases; however, we did strive to minimize the number of missed sepsis hospitalizations through repetitive chart reviews. Third, we did not include the data for patients readmitted to the hospital after more than 90 days of their initial admission and we did not examine all the risk factors for recurrent sepsis, including duration or type of antibiotics used. Fourth, we postulated that septic shock survivors had discharged successfully after they overcame initial infection. Therefore, we could not exclude possibilities about the remnant underlying problems related to sepsis including wrong choice, short duration of antibiotics or lack of source control. Lastly, because we did not have any data from outside our hospital, some recurrent sepsis readmissions at other hospitals may have been missed.

## 5. Conclusions

Our study demonstrated that patients with 90-day sepsis readmissions following previous hospitalization for septic shock were not uncommon (13.3%) and the rate of isolation of the same pathogen was 25% among those cases. Previous gram-negative bacteremia, and/or same site infection are predisposing factors for recurrent same-pathogen sepsis.

## Figures and Tables

**Figure 1 jcm-08-00181-f001:**
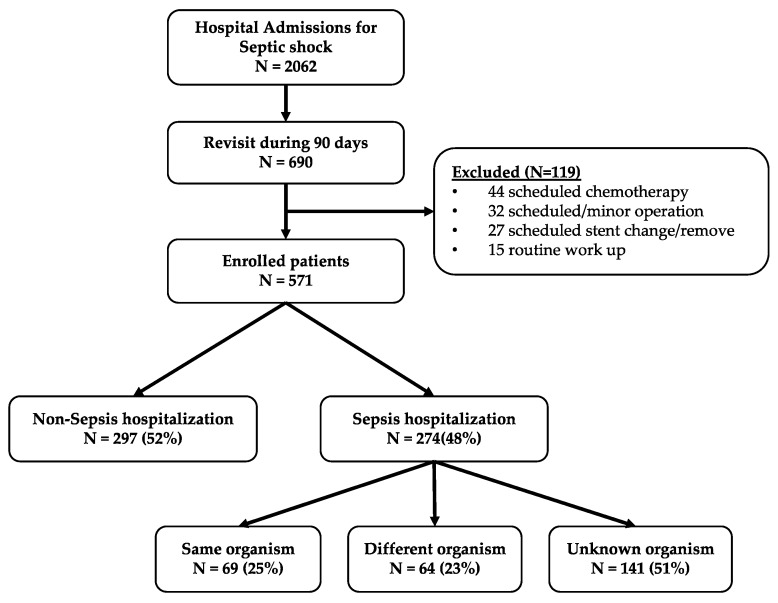
Flowchart of the study population.

**Table 1 jcm-08-00181-t001:** Characteristics of Patients with Recurrent Sepsis

Characteristics	Sepsis due to a Different Organism *n* = 64	Sepsis due to Unknown Organism *n* = 141	Sepsis due to the Same Organism *n* = 69	*p*-Value
Age	65 (59–72)	54 (57–71)	64 (68–72)	0.215
Male	38 (59.3)	92 (65.2)	31 (44.9)	0.027
Hypertension	22 (34.4)	46 (32.6)	31 (44.9)	0.308
Stroke	7 (10.9)	15 (10.6)	2 (2.9)	0.299
Diabetes	18 (28.1)	39 (27.7)	19 (27.5)	1.000
Coronary artery disease	11 (17.2)	19 (13.5)	5 (7.2)	0.478
Chronic pulmonary disease	13 (20.3)	37 (26.2)	13 (18.8)	0.357
Chronic renal failure	4 (6.3)	8 (5.7)	7 (10.1)	0.055
Liver cirrhosis	8 (12.5)	25 (17.7)	18 (26.1)	0.738
Malignancy	49 (76.6)	116 (82.3)	58 (84.1)	0.898
28 d mortality	2 (3.1)	3 (2.1)	1 (1.4)	1.000
90 d mortality	3 (4.7)	8 (5.7)	5 (7.2)	0.839
Mechanical ventilation	6 (9.4)	6 (4.3)	2 (2.9)	0.299
ICU stay	0 (0–3)	0 (0–2)	0 (0–3)	0.608
Same site	37 (57.8)	98 (69.5)	62 (89.9)	0.001

Data are presented as *n* (%) or median with interquartile ranges.

**Table 2 jcm-08-00181-t002:** Characteristics of Causal Organisms.

Characteristics	Total (%) *n* = 274	Different Organism (%) *n* = 64	Unknown Organism (%) *n* = 141	Same Organism (%) *n* = 69	*p*-Value
**Organism**					
Gram-positive	34 (12.4)	22 (34.3)	10 (7.1)	2 (2.9)	0.001
Gram-negative	181 (65.8)	46 (71.8)	69 (48.9)	66 (95.7)	<0.001
Viral	13 (4.7)	3 (4.7)	10 (7.1)	0 (0.0)	0.069
Fungi	4 (1.5)	1 (1.6)	2 (1.4)	1 (1.4)	1.000
**Site**					
Urinary	47 (17.1)	13 (20.3)	10 (7.1)	24 (34.8)	<0.001
Gastrointestinal	43 (15.7)	10 (15.6)	17 (12.1)	16 (23.2)	0.153
Hepatobiliary	103 (37.6)	22 (34.4)	61 (43.3)	20 (29.0)	0.153
Respiratory	72 (26.3)	25 (39.1)	44 (31.2)	3 (4.3)	0.165
Skin and soft tissue	10 (3.6)	4 (6.3)	4 (2.8)	2 (2.9)	0.638
Other	29 (10.6)	8 (12.5)	15 (10.6)	6 (8.7)	0.061

**Table 3 jcm-08-00181-t003:** Frequencies of Infectious Organisms in the Same Organism Group (*n* = 69).

Type	Organisms	Frequency (%)
Blood	*Staphylococcus aureus*	3 (8.1%)
	Hemolytic streptococci	1 (2.7%)
	*Escherichia coli*	19 (51.4%)
	*Klebsiella* spp.	6 (16.2%)
	*Enterobacteriaceae*	2 (5.4%)
	*Pseudomonas aeruginosa*	2 (5.4%)
	Other Gram-negatives	2 (5.4%)
	*Aspergillus* spp.	1 (2.7%)
	*Candida albicans*	1 (2.7%)
Urine	*Staphylococcus aureus*	1 (2.7%)
	Other Gram-positives	1 (2.7%)
	*Escherichia coli*	8 (21.6%)
	*Enterobacteriaceae.*	2 (5.4%)
	Pneumococci urine Ag	1 (2.7%)
Sputum	β-hemolytic *Streptococcus*	1 (2.7%)
	*Enterobacteriaceae*	1 (2.7%)
	*Pseudomonas aeruginosa*	1 (2.7%)
	*Acinetobacter baumannii*	2 (5.4%)
Pus	*Staphylococcus aureus*	1 (2.7%)
	*Escherichia coli*	4 (10.8%)
	*Klebsiella* spp.	1 (2.7%)
Stool	*Enterococcus* spp.	1 (2.7%)
	*Enterobacteriaceae*	1 (2.7%)

Abbreviations: spp. = species, Ag = antigen.

**Table 4 jcm-08-00181-t004:** Multivariate Analysis for Same Organism Reinfection in Study Patients

Variables	Multivariate Analysis		
	OR	95% CI	*p*-value
Male	0.589	0.295–1.176	0.133
Same site	6.894	2.390–19.886	<0.001
Gram-positive	0.194	0.037–1.016	0.052
Gram-negative	9.902	2.843–34.489	<0.001
UTI	4.331	1.723–10.882	0.002
Respiratory	0.312	0.086–1.133	0.077

Abbreviations: OR = odds ratio; CI = confidence interval; UTI = urinary tract infection.

**Table 5 jcm-08-00181-t005:** Performance Parameters for Predictors of Same Organisms in Study Patients

Variables	Sensitivity (%)	Specificity (%)	PPV (%)	NPV (%)	PLR	NLR
Same site	89.86 (80.21–100.00)	33.50 (30.00–40.00)	31.16 (28.54–33.90)	90.79 (82.64–95.33)	1.35 (1.19–1.53)	0.30 (0.15–0.63)
Gram-negative	95.65 (87.82–100.00)	44.17 (40.00–50.00)	36.46 (33.48–39.56)	96.81 (90.85–98.93)	1.71 (1.50–1.95)	0.10 (0.03–0.30)
UTI	34.78 (23.71–40.00)	88.83 (80.00–90.00)	51.06 (38.69–63.30)	80.26 (77.27–82.95)	3.12 (1.88–5.15)	0.73 (0.61–0.88)
Any of two factors	24.64 (15.05–30.00)	96.60 (80.00–100.00)	70.83 (51.26–84.87)	79.28 (76.94–81.45)	7.25 (3.14–16.74)	0.78 (0.68–0.89)
All three factors	95.65 (87.82–100.00)	55.34 (0.00–100.00)	41.77 (37.94–45.71)	97.44 (92.58–99.14)	2.14 (1.82–2.51)	0.08 (0.03–0.24)

Abbreviations: PPV = positive predictive value; NPV = negative predictive value; PLR = positive likelihood ratio; NLR = negative likelihood ratio; UTI = urinary tract infection. Data are presented with a 95% confidence interval.

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
