# Peer review of "Risk Factors for Same Pathogen Sepsis Readmission Following Hospitalization for Septic Shock"

_jcm, 2019, doi:10.3390/jcm8020181_

Reviewer 1 Report

Very nice study

I only have concern about your conclusion because it does not determine whether this same organism had a different anti-microbial sensitivity pattern

Do you know if these same organism infections were related to wrong antibiotic  or lack of source control or lack of correction of underlying problem related to infection such as residual urinary tract obstruction as an exmaple?

Perhaps your conclusion should be more along these lines rather than saying treat for the same organism?

Cheers

Author Response

Very nice study

I only have concern about your conclusion because it does not determine whether this same organism had a different anti-microbial sensitivity pattern. Do you know if these same organism infections were related to wrong antibiotic or lack of source control or lack of correction of underlying problem related to infection such as residual urinary tract obstruction as an exmaple? Perhaps your conclusion should be more along these lines rather than saying treat for the same organism?

Response: Thank you for your generous comment. We totally agree with your concern about conclusion. We postulated that septic shock survivors had discharged successfully after they overcame infections. Regretfully, there were no way to know with our registry about the influence of wrong choice of antibiotics, short duration of antibiotics regimen, lack of source control, or lack of correction of underlying problems. In addition, we reviewed anti-microbial sensitivity data of all 69 same organism. Among these, 9 pathogens (13%; E. coli= 5; Klebsiellaspecies= 2; Enterobacter= 1; Pseudomonas= 1) became resistant and rest pathogens had same susceptibilities of previous antibiotics. However, it cannot be conclusive for providing guideline of empirical antibiotics because of relatively small sample size. Therefore, we have added limitations and revised our conclusions as below per your suggestion.

“Among 69 same organism patients, 9 pathogens (13%; E. coli= 5; Klebsiellaspecies= 2; Enterobacter= 1; Pseudomonas= 1) became resistant and rest pathogens had same susceptibilities of previous antibiotics. However, it cannot be conclusive for providing guideline of empirical antibiotics because of relatively small sample size in our study.” (page 7, line 292 – 295)

“Fourth, we postulated that septic shock survivors had discharged successfully after they overcame initial infection. Therefore, we could not exclude possibilities about the remnant underlying problems related to sepsis including wrong choice, short duration of antibiotics, or lack of source control.” (page 8, line 335-338)

“Our study demonstrated that patients with 90-day sepsis readmissions following previous hospitalization for septic shock were not uncommon (13.3%), and the rate of isolation of the same pathogen was 25% among those cases. Based on our results, we suggest that for readmitted sepsis patients with previous gram-negative bacteremia, urinary tract infection, and/or same site infection after hospitalization the treating physician may initiate treatment with antibiotics for recurring or relapsing of the same infectious pathogen. Previous gram-negative bacteremia, and/or same site infection are predisposing factors for recurrent same-pathogen sepsis.” (page 8, line 343 - 344)

Reviewer 2 Report

The authors proposed a retrospective analysis of patients re-admitted in hospital for infections after a previous admission for septic shock with the aim of identifying risk factors for relapse of infections by the same micro-organisms that may guide empiric antibiotic therapy. They concluded that with ‘previous gram-negative bacteremia, urinary tract infection, and/or same site infection after hospitalization the treating physician may initiate treatment with antibiotics for recurring or relapsing of the same infectious pathogen’. The idea is suggestive but unfortunately, the authors did not provide any information on antibiotic susceptibility and, thus, the conclusion can be not appropriate and, indeed, unsafe. As well known, recurrent infections by the same microorganism, particularly infections by Enterobacteriaceae, are usually resistant (become resistant) to antibiotics used for the treatment of the first episode. The clarification of this point by providing antibiograms or at least changes in antibiotic susceptibility is mandatory. More, this point should be well discussed in the manuscript.

ABSTRACT: the aim of the abstract does not match with aim described in the main text

METHODS:

Lines 83: The authors decide to include in different microorganisms group patients with negative culture. This procedure is not further justified. I guess may be more appropriate to include this population in a specific different group.

lines 88-89: I suggest do not use the term primary outcome but a more general outcome. Furthermore, I suggest listing all the outcome measured (better in section 2.3 Data analysis)

RESULTS

Lines 144-148: The methods used for multivariable analysis should be moved in statistical (or data) analysis section and should be better clarified (selection of variables and robustness of the model)

Author Response

The authors proposed a retrospective analysis of patients re-admitted in hospital for infections after a previous admission for septic shock with the aim of identifying risk factors for relapse of infections by the same micro-organisms that may guide empiric antibiotic therapy. They concluded that with ‘previous gram-negative bacteremia, urinary tract infection, and/or same site infection after hospitalization the treating physician may initiate treatment with antibiotics for recurring or relapsing of the same infectious pathogen’. The idea is suggestive but unfortunately, the authors did not provide any information on antibiotic susceptibility and, thus, the conclusion can be not appropriate and, indeed, unsafe. As well known, recurrent infections by the same microorganism, particularly infections by Enterobacteriaceae, are usually resistant (become resistant) to antibiotics used for the treatment of the first episode. The clarification of this point by providing antibiograms or at least changes in antibiotic susceptibility is mandatory. More, this point should be well discussed in the manuscript.

Response: Thank you for your generous comment. We totally agree with your concern about conclusion. We reviewed anti-microbial sensitivity data of all 69 same organism. Among these, 9 pathogens (13%; E. coli= 5; Klebsiellaspecies= 2; Enterobacter= 1; Pseudomonas= 1) became resistant and rest pathogens had same susceptibilities of previous antibiotics. However, it cannot be conclusive for providing guideline of empirical antibiotics because of relatively small sample size. Therefore, we have added limitations and revised our conclusions as below per your suggestion.

“Among 69 same organism patients, 9 pathogens (13%; E. coli= 5; Klebsiellaspecies= 2; Enterobacter= 1; Pseudomonas= 1) became resistant and rest pathogens had same susceptibilities of previous antibiotics. However, it cannot be conclusive for providing guideline of empirical antibiotics because of relatively small sample size in our study.” (page 7, line 292 – 295)

“Our study demonstrated that patients with 90-day sepsis readmissions following previous hospitalization for septic shock were not uncommon (13.3%), and the rate of isolation of the same pathogen was 25% among those cases. Based on our results, we suggest that for readmitted sepsis patients with previous gram-negative bacteremia, urinary tract infection, and/or same site infection after hospitalization the treating physician may initiate treatment with antibiotics for recurring or relapsing of the same infectious pathogen.Previous gram-negative bacteremia, and/or same site infection are predisposing factors for recurrent same-pathogen sepsis.” (page 8, 343 - 344)

ABSTRACT: the aim of the abstract does not match with aim described in the main text

Response: As your previous comment, we revised our suggestion about guideline of empirical antibiotics. Our study showed that gram-negative bacteremia, and/or same site infection are risk factors for recurrent same-pathogen sepsis. And this could provide direction of future study about antibiotic susceptibility of recurrent pathogen. However, because of small sample size of our study and limitation of information about previous antibiotic usage or duration, we have omitted our conclusion and matched result of main text with abstract.

METHODS:

Lines 83: The authors decide to include in different microorganisms group patients with negative culture. This procedure is not further justified. I guess may be more appropriate to include this population in a specific different group.

Response: Thank you for your valuable comments. We accepted your opinion and included culture-negative organism group in a separate “unknown organism” group. We changed the sentence in method (2.2 Data collection and definition, 2.3. Statistical analyses, 3. Results), Figure 1, Table 1, and Table 2 and previous version of Table 1 and 2 were provided as Supplement 1 and 2. Culture-negative septic shock was the major limitation of the study related to pathogen identification. Previous study reported that the respiratory and intraabdominal systems were the most frequent sites of infection in culture-negative septic shock. In our study, among 141 patients of culture-negative septic shock, 72 (26.3%) were the respiratory and 146 (53.3%) were intraabdominal origin. However, urinary tract infection, gram-negative bacteremia, and same site of infection were still independent risk factor for readmission within 90 days.

“When all the cultures were negative (including those in the first and second admissions) reflecting no definite growth of any specimen, we included these cases into the “unknown organism” group.” (page 2, line 85-86)

“The Kruskal-Wallis test was used for comparisons between same, different, unknown sepsis organism groups.” (page 3, line 98-99)

“We found no age-related differences between the three groups. Male was more common in unknown group. Malignancy was the most common underlying disorder followed by hypertension. We did not observe significant differences in previous medical conditions such as hypertension, stroke, diabetes, coronary artery disease, chronic pulmonary disease, liver cirrhosis, chronic renal failure, or malignancy between the groups. Over the entire study period, 28- and 90-day mortalities, mechanical ventilation requirements, and length of intensive care unit stays were statistically similar in three groups. However, we found the same site of reinfection to be significantly more frequent in the “same organism” group than in the “different organism” and “unknown organism” group (79.7% vs 62.5 vs 79.7%; p < 0.001).” (page 3, line 124-152)

“The most common category of organism causing repetitive sepsis was the gram-negatives (n = 180; 65.7%), meanwhile, gram-positive infections were not as common (n = 34; 12.4%) (Table 2). Identical gram-negative species caused recurrent sepsis were predominant in same group (95.7% vs 71.8% vs 48.9%, respectively). The infection sites in cases of sepsis readmission included hepatobiliary (37.6), gastrointestinal (15.7%), respiratory (26.3%), and urinary tract (17.1%). Culture-negative samples comprised 141 (51.4%) of the total and the respiratory tract (n = 44; 31.2%) and intraabdominal system (n = 146; 53.3%) were common. In urinary system, same pathogen was isolated more often than others (34.8% vs 20.3% vs 7.1%, respectively).” (page 4, line 157 – 164)

“Because of the uncertainty, we considered the culture negative as “different organism” group for the sensitive analysis and the trend of the results were similar (Supplement 1, 2), and with this approach we observed 75% new, different infections and 25% same persistent or relapsing pathogens.” (page 7, line 296 - 299)

Table 1.Characteristics of Patients with Recurrent Sepsis

Characteristics

Sepsis due to a different organism

n = 64

Sepsis due to unknown organism

n = 141

Sepsis due to the same organism

n = 69

p-value

Age

65 (59 – 72)

54 (57– 71)

64 (68 – 72)

0.215

Male

38 (59.3)

92 (65.2)

31 (44.9)

0.027

Hypertension

22 (34.4)

46 (32.6)

31 (44.9)

0.308

Stroke

7 (10.9)

15 (10.6)

2 (2.9)

0.299

Diabetes

18 (28.1)

39 (27.7)

19 (27.5)

1.000

Coronary artery disease

11 (17.2)

19 (13.5)

5 (7.2)

0.478

Chronic pulmonary disease

13 (20.3)

37 (26.2)

13 (18.8)

0.357

Chronic renal failure

4 (6.3)

8 (5.7)

7 (10.1)

0.055

Liver cirrhosis

8 (12.5)

25 (17.7)

18 (26.1)

0.738

Malignancy

49 (76.6)

116 (82.3)

58 (84.1)

0.898

28 d mortality

2 (3.1)

3 (2.1)

1 (1.4)

1.000

90 d mortality

3 (4.7)

8 (5.7)

5 (7.2)

0.839

Mechanical ventilation

6 (9.4)

6 (4.3)

2 (2.9)

0.299

ICU stay

0 (0 – 3)

0 (0 – 2)

0 (0 – 3)

0.608

Same site

37 (57.8)

98 (69.5)

62 (89.9)

0.001

Data are presented as n (%) or median with interquartile ranges.

Table 2.Characteristics of Causal Organisms

Characteristics

Total (%)

n = 274

Different organism (%)

n = 64

Unknown organism (%)

n = 141

Same organism (%)

n = 69

p-value

Organism

  Gram-positive

34 (12.4)

22 (34.3)

10 (7.1)

2 (2.9)

0.001

  Gram-negative

181 (65.8)

46 (71.8)

69 (48.9)

66 (95.7)

< 0.001

  Viral

13 (4.7)

3 (4.7)

10 (7.1)

0 (0.0)

0.069

  Fungi

4 (1.5)

1 (1.6)

2 (1.4)

1 (1.4)

1.000

Site

  Urinary

47 (17.1)

13 (20.3)

10 (7.1)

24 (34.8)

< 0.001

  Gastrointestinal

43 (15.7)

10 (15.6)

17 (12.1)

16 (23.2)

0.153

  Hepatobiliary

103 (37.6)

22 (34.4)

61 (43.3)

20 (29.0)

0.153

  Respiratory

72 (26.3)

25 (39.1)

44 (31.2)

3 (4.3)

0.165

  Skin and soft tissue

10 (3.6)

4 (6.3)

4 (2.8)

2 (2.9)

0.638

  Other

29 (10.6)

8 (12.5)

15 (10.6)

6 (8.7)

0.061

lines 88-89: I suggest do not use the term primary outcome but a more general outcome. Furthermore, I suggest listing all the outcome measured (better in section 2.3 Data analysis)

Response: As your suggestion, we changed the term “primary outcome” to “main clinical outcome” and moved in section 2.3. Statistical analyses.

“The main clinical outcome was finding the rate and risk factors of readmission due to sepsis caused by the “same organism” within 90 days of discharge following treatment for septic shock.” (page 3, line 111 - 112)

RESULTS

Lines 144-148: The methods used for multivariable analysis should be moved in statistical (or data) analysis section and should be better clarified (selection of variables and robustness of the model)

Response: We moved sentences “and after using the process of forward-backward stepwise regression” from 3.2. Etiology of recurrent sepsisto 2.3. Statistical analyses.

“To identify risk factors for the identical pathogen, we included variables with an entry-level significance of p < 0.05 in the univariate analysis in a stepwise multivariate analysis, and after using the process of forward-backward stepwise regression, we reported the results as odds ratios (ORs) and 95% confidence intervals (CIs).” (page 3, line 95-98)

“In the multivariable model, three clinical factors were independent risk factors for infections by the same organism: same site (OR, 6.894; 95% CI, 2.390–19.886; p < 0.001), gram-negative (OR, 9.902; 95% CI, 2.843–34.489; p < 0.001), and UTI (OR, 4.331; 95% CI, 1.723 – 10.882; p = 0.002).” (page 5, line 146-149)

Reviewer 3 Report

Very useful study, but some considerations:

- In the baseline characteristics, do you consider or collect info about patient who are chronic microorganism carries (i.e. patients with bronchiectasis or any other chronic respiratory disease)

- Could be possible to evaluate if the "same organism" had the same anti-microbial sensitivity in the first and second admission? because if this aspect is different, maybe you would be talking about different strains instead of different microorganisms within this group, and maybe the treatment in the first admission it was not totally correct.

- And about the treatment, could be possible to analize the adequacy of treatments in the first admission and its implications in the readmission?

Because this study is a retrospective cohort, maybe some of this considerations could be difficult to answer, but they are limitations to add to the discussion

Author Response

Very useful study, but some considerations:

- In the baseline characteristics, do you consider or collect info about patient who are chronic microorganism carries (i.e. patients with bronchiectasis or any other chronic respiratory disease)

Response: In table 1, we had baseline characteristics of chronic pulmonary disease such as bronchiectasis, chronic obstructive pulmonary disease, or interstitial lung disease. (page 4, table 1)

- Could be possible to evaluate if the "same organism" had the same anti-microbial sensitivity in the first and second admission? because if this aspect is different, maybe you would be talking about different strains instead of different microorganisms within this group, and maybe the treatment in the first admission it was not totally correct.

Response: Thank you for your generous comment. We totally agree with your concern about conclusion. We reviewed anti-microbial sensitivity data of all 69 same organism. Among these, 9 pathogens (13%; E. coli= 5; Klebsiellaspecies= 2; Enterobacter= 1; Pseudomonas= 1) became resistant and rest pathogens had same susceptibilities of previous antibiotics. However, it cannot be conclusive for providing guideline of empirical antibiotics because of relatively small sample size. Therefore, we have added limitations and revised our conclusions as below per your suggestion.

“Among 69 same organism patients, 9 pathogens (13%; E. coli= 5; Klebsiellaspecies= 2; Enterobacter= 1; Pseudomonas= 1) became resistant and rest pathogens had same susceptibilities of previous antibiotics. However, it cannot be conclusive for providing guideline of empirical antibiotics because of relatively small sample size in our study.” (page 7, line 292 – 295)

“Our study demonstrated that patients with 90-day sepsis readmissions following previous hospitalization for septic shock were not uncommon (13.3%), and the rate of isolation of the same pathogen was 25% among those cases. Based on our results, we suggest that for readmitted sepsis patients with previous gram-negative bacteremia, urinary tract infection, and/or same site infection after hospitalization the treating physician may initiate treatment with antibiotics for recurring or relapsing of the same infectious pathogen.Previous gram-negative bacteremia, and/or same site infection are predisposing factors for recurrent same-pathogen sepsis.” (page 8, 343 - 344)

- And about the treatment, could be possible to analize the adequacy of treatments in the first admission and its implications in the readmission?

Response: We postulated that septic shock survivors had discharged successfully after they overcame infections. Regretfully, there were no way to know with our registry about the influence of wrong choice of antibiotics, short duration of antibiotics regimen, lack of source control, or lack of correction of underlying problems. Therefore, we have added limitations and revised our conclusions as below per your suggestion.

“Fourth, we postulated that septic shock survivors had discharged successfully after they overcame initial infection. Therefore, we could not exclude possibilities about the remnant underlying problems related to sepsis including wrong choice, short duration of antibiotics, or lack of source control.” (page 8, line 335-338)

Because this study is a retrospective cohort, maybe some of this considerations could be difficult to answer, but they are limitations to add to the discussion

Round  2

Reviewer 2 Report

No further comments